# CASD: Enhancing Generation Accuracy via Context-Aware Speculative Decoding

## Abstract

With recent advancements in long-context model variants, Large language models (LLMs) can conveniently process different types of task-related information by simply converting them into an input sequence, even consisting of over 100K tokens. Though with a simple and unified form, there is still considerable room in leveraging input context effectively and efficiently. In this paper, we propose a simple yet effective CASD (**C**ontext-**A**ware **S**peculative **D**ecoding) method to boost context usage. CASD is a decoding algorithm that requires no extra training or draft models. It improves not only generation performance but also inference efficiency. Experiments on **8** datasets (including question answering, summarization and code completion tasks in LongBench) show that CASD increases the average generation score by **3.3** points. CASD achieves a mean acceptance length of 3.10 and a speed-up ratio of **1.99**. Moreover, CASD integrates effectively with context compression technology, addressing the issue of excessive memory overhead caused by long contexts. Since CASD directly retrieves token-level content from the input context to boost the generation accuracy, it can effectively mitigate the possible side-effects of context compression methods when crucial context information is dropped. Our anonymous code is available at https://anonymous.4open.science/r/CASD.

## 1 Introduction

Directly prompting large language models (LLMs) with all relevant task information (e.g., task description, query, relevant documents) has become a common setting when serving LLMs for real-world applications, especially with recent long-context models (Anthropic, 2024; Meta-AI, 2024). However, overly long input context might pose great challenges to LLMs in accurately capturing salient context information and achieving decent generation performance, possibly making LLMs unfollow instruction and output hallucinations (Belyi et al., 2024). Besides, since prefilling long context input significantly increases memory overhead, many recent works explore prompt compression methods (Jiang et al., 2023; Xu et al., 2024; Pan et al., 2024) to lower inference costs. These compression techniques will inevitably drop part of critical context information, leading to inferior generation performance.

Speculative decoding (Stern et al., 2018; Leviathan et al., 2023; Cai et al., 2024; Li et al., 2024) algorithms have recently shown excellent performance in lossless decoding acceleration for autoregressive models. They generate multiple tokens in a single step by drafting with a small model and verifying with the target model. The draft process can act as an interface to access external information. REST (He et al., 2024) retrieves drafts from a general database. However, the input context can provide drafts that are more relevant to the query and of higher quality. Inspired by this, we propose a context-aware speculative decoding algorithm with conditional verification to enhance the generation accuracy and efficiency of LMs, which is abbreviated as CASD. The overall design is demonstrated in Figure 1. CASD uses the original context as the draft source and replaces the strict verification mode in conventional speculative decoding with conditional verification, allowing the model to access and utilize the retrieved information for accurate generation. In each decoding step, pieces of the relevant context are verified and accepted at the token level, thus making better use of the contextual information.

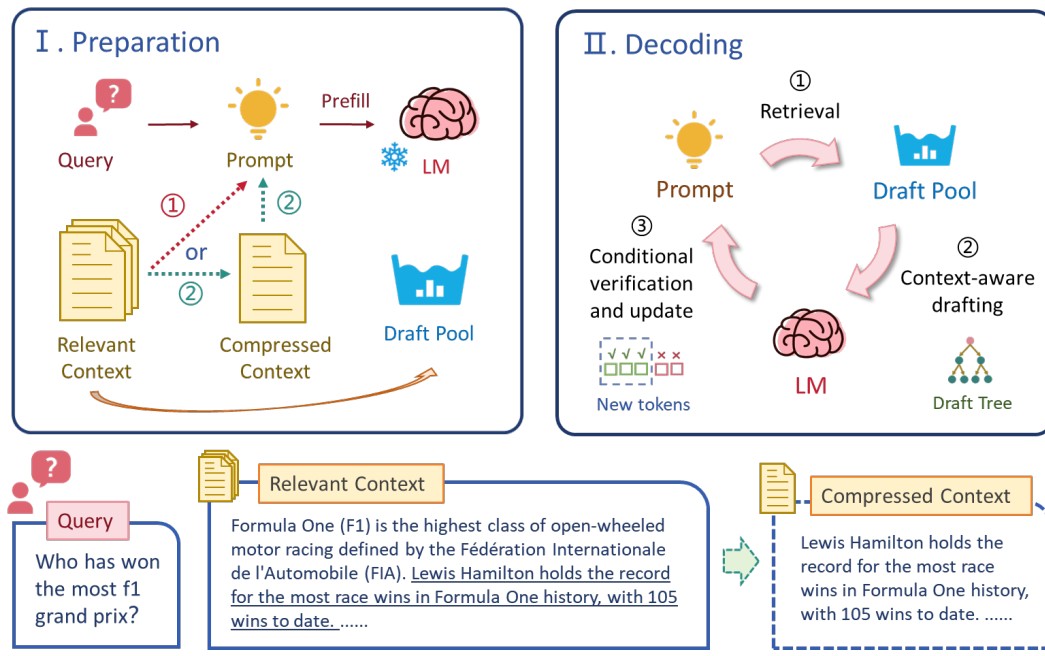

Figure 1: Overall illustration of CASD. CASD constructs a draft pool with the full relevant context. Then, the context can be directly inserted into the prompt or be compressed to adapt to the maximum input length determined by the model or memory. After prefilling the prompt into the fixed LM, CASD retrieves a draft tree from the draft pool according to the current prefix. The fixed LM performs parallel conditional verification on the draft tree based on the output distribution. One or multiple tokens are accepted in each decoding step.

Experimental results in Section 4.2 show that CASD outperforms the baseline for 8 datasets in question answering, summarization and code completion tasks. With a manually adapted verification condition, the performance of CASD can be further improved, demonstrating great potential for reusing the context at the decoding stage. Experiments in Section 4.4 indicate that CASD can be well adapted to prompt compression techniques and can make up for the performance loss caused by the compression to some degree. CASD improves the average scores by 2.09, 1.00 and 1.39 under a compression ratio of $3\times$, $5\times$ and $10\times$ when conducting prompt compression by LLMLingua-2 (Pan et al., 2024). As an external component, the draft pool does not need to be input into the model, so it has no strict length limit. For context within 1M tokens, the draft retrieval time is less than 0.01 seconds. As a result, we can construct the draft pool with full context while inputting the compressed one. Even if some crucial passages were cut out, the draft pool retains the possibility that the LM considers them. Besides, CASD achieves a speed-up ratio of up to 2 times compared to autoregressive decoding, which is detailed in Section 5.1.

To sum up, we introduce a simple yet effective approach called CASD in this paper, which makes better use of the relevant context during generation. Our contributions are as follows:

• We propose a novel decoding paradigm CASD for LMs, which does not require additional training and draft models. It applies to most scenarios where each request contains a question and the relevant context. CASD reuses the input context during the decoding process. It enhances generation accuracy from the token level, thus making better use of the relevant context.

• The proposed method can be effectively combined with prompt compression methods. While dealing with the problems caused by long context input, it also alleviates the loss in performance caused by the compression of vital information.

• CASD improves generation accuracy while reducing inference latency. It increases the average score on 8 datasets by 3.3 points and increases throughput by up to 2 times.

## 2 RELATED WORK

**Long Context Generation** Using context to improve generation performance is typical in retrieval-augmented generation (RAG) scenarios. RAG methods (Zheng et al., 2023; Dai et al., 2023; Gao et al., 2023; Fan et al., 2024; Zhao et al., 2024) retrieves relevant documents based on the given input for model reference, thus improving the quality of generation. The naive approach (Ma et al., 2023) applies the search engine as the retriever and directly combines the retrieved documents with the user query as the input for frozen LLMs (e.g., GPT-4 (Achiam et al., 2023)). Most RAG methods (Yoran et al., 2023; Luo et al., 2023; Asai et al., 2023; Melz, 2023; Yan et al., 2024; Wang et al., 2024b) leveraging context on the input side to enhance generation. Self-RAG (Asai et al., 2023) introduces generating reflection tokens to enable customizing models' behaviors for different tasks. Speculative RAG (Wang et al., 2024b) adopts instruct-tuned draft models to drafting according to different retrieved documents and uses the target model to pick out the best draft as the final response. Apart from them, CoG Lan et al. (2023) proposes a encoder-based model architecture to seek suitable text spans from the context during generation. Cao et al. (2024) improves CoG through linguistic heuristics initialization and iterative self-reinforcement.

**Prompt compression** (Li et al., 2023; Jiang et al., 2023; Xu et al., 2024; Pan et al., 2024) methods have been proposed to extract crucial information from context and deal with the burden of long text on models and hardware resources. LLM-Lingua (Jiang et al., 2023) employs a small language model to conduct iterative token-level prompt compression, which takes into account the conditional dependencies between tokens. LLM-Lingua2 (Pan et al., 2024) treats this problem as a binary classification problem and trains an encoder model as a compressor.

**Speculative decoding** (SD) (Stern et al., 2018; Leviathan et al., 2023; Xia et al., 2023; 2024) adopts a "drafting-verification" pattern to lossless accelerates autoregressive decoding. At each decoding step, a small model is used to draft the following few tokens based on the current input. Then, the target model verifies the draft in parallel and accepts tokens that are consistent with the original output. In this way, multiple tokens can be generated in a single step. Some works (Leviathan et al., 2023; Cai et al., 2024; Li et al., 2024) employ independent small models or additional trained modules as the draft models, while others (Saxena, 2023; Fu et al., 2024; He et al., 2024) retrieve drafts from a draft pool. The draft structure has evolved from n-grams (Leviathan et al., 2023; Fu et al., 2024) to draft trees (Cai et al., 2024; Li et al., 2024; Wang et al., 2024a). Among them, REST (He et al., 2024) uses a common public data source to build a draft pool and retrieves the tree structure draft according to the last several tokens of the current sequence at each decoding step.

## 3 CASD

In this section, we introduce CASD, a context-aware speculative decoding algorithm based on conditional verification that improves generation performance and efficiency. Our approach applies to scenarios where the input can be separated into a query and a supporting context.

Given a query $q$ and the relevant context $d_q((q, d_q) \in \mathbb{D})$ , where $\mathbb{D}$ is the whole dataset, our goal is to make the language model $M$ generate an answer $\hat{A}_q$ that are as close to the ground truth $A_q$ as possible based on $q$ and $d_q$. Query $q$ and $d_q$ are usually highly correlated. In addition, $A_q$ often contains some the original phrases in $d_q$.

Based on the above observations, we argue that the dependence of the content generated by the model on the relevant context can be enhanced to achieve better generation accuracy. We design CASD, a context-aware speculative decoding algorithm with conditional verification. By retrieving high-quality drafts from the relevant context according to the current prompt, CASD enables the LM to call the original fragment in the context directly under certain conditions. It enhances generation performance at a token level.

The overall design of CASD is demonstrated in Algorithm 1. CASD does not require any training or external data other than the context that come with the question. Specifically, we construct a draft pool with $d_q$ for each request. Query $q$ and $d_q$ are applied to a task-related template to construct a prompt $p = (x_1, x_2, ..., x_l)$, where $x_i(i = 1, 2, ..., l)$ represents each token and $l$ represents the prefix length, which is then input into the model to obtain the first next token:

$$y_{l+1} \sim p(x_{l+1}|(x_1, x_2, ..., x_l), M), \tag{1}$$

---

**Algorithm 1** Context-Aware Speculative Decoding.

---

**Input:**      Query $\boldsymbol{q}$, relevant context $\boldsymbol{d_q}$, LM $M$, threshold $\delta$.
**Output:**      Prediction $\hat{A}_{\boldsymbol{q}}$
 1: $R \leftarrow \boldsymbol{d_q}$                           ▷ Initialize the draft retriever $R$ with full context.
 2: **if** compress prompt **then**
 3:     $\boldsymbol{d_q} \leftarrow Compress(\boldsymbol{d_q})$
 4: **end if**
 5: $\boldsymbol{p} = (x_1, x_2, ..., x_l) \leftarrow \boldsymbol{q} + \boldsymbol{d_q}$
 6: **while** $\langle eos \rangle$ not in $\boldsymbol{p}$ **do**
 7:     $\mathbb{T} \leftarrow R(\boldsymbol{p})$                ▷ Retrieve the draft tree according to the current prompt.
 8:     $\mathbb{P} \leftarrow M(\mathbb{T})$            ▷ Get the next token distributions for all tokens in the draft.
 9:     $\boldsymbol{y} \leftarrow Conditional\text{-}verify(\mathbb{P}, \mathbb{T}, \delta)$     ▷ Accept the longest draft that meets the conditions.
10:     $\boldsymbol{p} \leftarrow \boldsymbol{p} + \boldsymbol{y}$
11: **end while**
12: $\hat{A}_{\boldsymbol{q}} \leftarrow \boldsymbol{p}$

---

where $p(\cdot | \cdot, M)$ represents the output probability distribution of $M$.

**Context-aware drafting** We conduct drafting at each decoding step. Take the first decoding step as an example. The draft pool provides a draft tree $\mathbb{T}$ and the corresponding tree attention mask according to the current prompt $\boldsymbol{p} = (x_1, x_2, ..., x_l, y_{l+1})$. $\mathbb{T}$ is defined as:

$$\mathbb{T} = (\mathbb{V}, \mathbb{E}), \mathbb{V} = \bigcup_{i=l+1}^{l+m} \bigcup_{j=1}^{n_i} \left\{ \hat{y}_i^j \right\}, \tag{2}$$

where $\mathbb{V}$ and $\mathbb{E}$ is the set of its nodes and edges. $n_i$ is the number of retrieved tokens in the $i_{th}$ layer of $\mathbb{T}$. $m$ is the depth of $\mathbb{T}$. We use the exact-match-based retrieval algorithm proposed by He et al. (2024) to retrieve and construct the draft tree from the draft pool. The tree size changes as the retrieval result changes. Then we input $\mathbb{T}$ and the attention mask into the model and get the output probability distribution $p(y_i | \mathbb{F}(\hat{y}_i^j), M)$ for each node, where $\mathbb{F}(\hat{y}_i^j)$ is the set of all parent nodes (including prefix) of $\hat{y}_i^j$.

**Conditional verification** For standard speculative decoding, tokens in $\mathbb{T}$ are verified by position order. For draft token $\hat{y}_i^j$, it is accepted if it matches the token sampled from $p(y_i | \mathbb{F}(\hat{y}_i^j), M)$. If $\hat{y}_i^j$ is rejected, all its child nodes will be rejected. Through this rigorous verification, the output of the model at any temperature will be consistent with the vanilla autoregressive decoding. However, our goal is to use the relevant context to enhance the reliability of the generated content. An overly strict verification mode limits model utilization of key information in the relevant context, while over-reliance on the context may reduce the quality of the output. Therefore, we set a probability threshold $\delta$ to balance the model's confidence in the context and the quality of generation. Draft token $\hat{y}_i^j$ will be accepted under the following condition:

$$p(\hat{y}_i^j | \mathbb{F}(\hat{y}_i^j), M) > max(\delta, p(\langle eos \rangle | \mathbb{F}(\hat{y}_i^j), M)). \tag{3}$$

The probability of $\langle eos \rangle$ token is taken into consideration to avoid duplicate answers generation. Note that its parent nodes should still have been accepted. Otherwise, it will be rejected unconditionally. In addition, to further ensure the quality of generation, we also consider a mixed condition verification mode, which accepts $\hat{y}_i^j$ if:

$$p(\hat{y}_i^j | \mathbb{F}(\hat{y}_i^j), M) > max(\delta, p(\langle eos \rangle | \mathbb{F}(\hat{y}_i^j), M)) \quad and \quad \hat{y}_i^j \in Topk(p(y_i | \mathbb{F}(\hat{y}_i^j), M)), \tag{4}$$

where $Topk(p)$ return the set of $k$ tokens with the highest probability in distribution $p$.

We argue that using the probability threshold condition is superior to using the top-k condition alone since the output probability directly reflects the model's confidence in each draft token. Moreover, the verification condition of top-k is not applicable for some extreme cases. For example, suppose the probability of a certain token is close to 1, and the probability of all other words is close to 0, except for the token with the highest probability. In that case, no other token should be accepted. If all tokens have the same probability, then they should all be accepted or rejected at the same time. However, the top-k verification condition will only pass individual tokens and reject others.

**Combination with prompt compression** Excessively long inputs have been a challenge for computing resources and a maximum length of some LLMs, which also affect inference efficiency. However, this is not a problem for the plug-in draft pool. For context with less than 1M tokens, changes in context length have little impact on the retrieval efficiency of the draft pool, which is elaborated in Section 5.1. To cope with the challenges posed by long context issue, we construct the draft pool with full context and input the compressed prompt by LLMLingua-2 Pan et al. (2024). Note that we only compress the context and leave the original question or instruction intact. In this way, we retain the potential for missing critical information caused by prompt compression to influence the model's output.

# 4 EXPERIMENTS

## 4.1 SETTINGS

We compare the performance of CASD with vanilla autoregressive decoding with LLaMA3.1-8B-Instruct (Meta-AI, 2024). We did not compare with the similar enhanced generation methods because they require additional training or use additional models. We evaluate the proposed method on 3 different tasks: 1) **Question Answering (QA)**: Nature Question (NQ) (Kwiatkowski et al., 2019), TriviaQA (TQA) (Joshi et al., 2017), 2WikiMQA (Ho et al., 2020) and HotpotQA (Yang et al., 2018), 2) **Summarization**: Multi-News (Fabbri et al., 2019) and GovReport (Huang et al., 2021), 3) **Code Completion**: RepoBench-P (Liu et al., 2024) and LCC (Guo et al., 2023). For the NQ dataset, we randomly sampled 300 queries from the validation set for evaluation. We use subsets sampled by LongBench (Bai et al., 2023) for other datasets. We report F1 score for QA datasets, ROUGE-L for summarization datasets and Edit Sim for code completion datasets.

We show the performance of CASD with two different verification strategies: 1) **CASD (Fixed)**: We set the threshold $\delta$ to 0.1 for all experiments, 2) **CASD (Mixed)**: With the threshold set to 0.1, the probability of all accepted tokens during verification is required to rank in the top 5 in their corresponding output distributions. In addition, to explore the upper limit of CASD, we conduct two other sets of experiments with manual parameter adjustment: 1) **CASD (Oracle-D)** (Dataset-level threshold adjustment): We select the best threshold $\delta$ in {0.1, 0.01, 1e-3, 1e-4, 1e-5} for each dataset (without the top-k requirement). 2) **CASD (Oracle-S)** (Sample-level threshold adjustment): Similarly, we tune $\delta$ for each sample within the same search space. We did not consider the lower thresholds as the coherence of the model's responses will be significantly diminished.

## 4.2 MAIN RESULTS

Table 1: Performance of different methods with LLaMA3.1-8B-Instruct. For all metrics, higher scores indicate better performance. The best results are in bold (except for Oracle-D and Oracle-S).

| Method | NQ | TQA | 2WikiMQA | HotpotQA | Multi-News | GovReport | RepoBench-P | LCC | Avg. |
|---|---|---|---|---|---|---|---|---|---|
| | F1 | F1 | F1 | F1 | ROUGE-L | ROUGE-L | Edit Sim | Edit Sim | |
| Vanilla | 29.13 | 92.13 | 42.83 | 49.80 | 13.51 | 15.96 | 48.11 | 66.05 | 44.69 |
| **CASD (Fixed)** | 33.51 | **92.62** | **44.67** | 51.72 | **14.97** | **20.09** | 56.31 | 69.24 | 47.89 |
| **CASD (Mixed)** | **33.69** | **92.62** | 43.95 | **52.46** | 14.83 | 20.06 | **56.63** | **69.46** | **47.96** |
| **CASD (Oracle-D)** | 43.78 | 92.75 | 44.52 | 52.46 | 15.14 | 20.13 | 58.08 | 70.72 | 49.70 |
| **CASD (Oracle-S)** | 53.84 | 92.75 | 45.43 | 53.84 | 17.79 | 23.40 | 61.14 | 74.16 | 52.79 |

Table 1 shows the performance of CASD on the above tasks. We report the average scores of the 8 datasets in the last column. Even with a fixed heuristic threshold, CASD can effectively improves the generation performance. CASD (Fixed) and CASD (Mixed) outperform the baselines on all datasets. CASD (Mixed) is slightly better than CASD (Fixed) in terms of the average score. It improves the score from 44.69 to 47.96 compared to vanilla decoding.

Furthermore, the results of CASD (Oracle-D) and CASD (Oracle-S) highlight the high-performance potential of CASD. If equipped with appropriate thresholds, the generation quality can be further improved, especially for NQ, where the maximum improvement is from 29.13 (baseline) to 53.84.

## 4.3 PERFORMANCE UNDER DIFFERENT THRESHOLDS

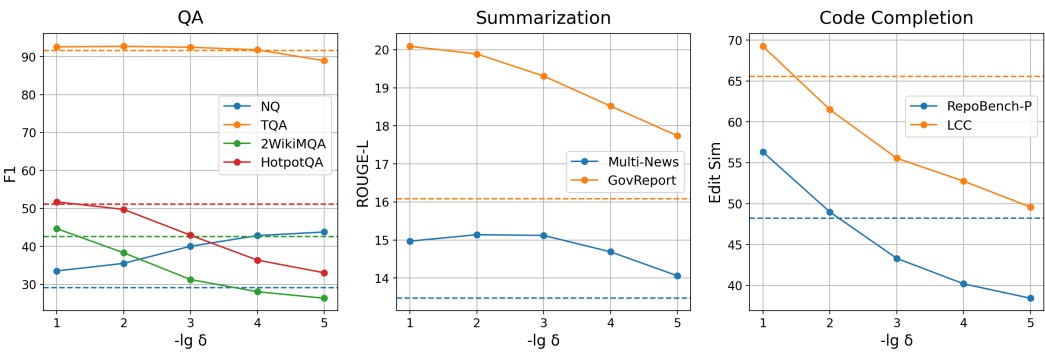

Figure 2: Performance of CASD under different thresholds. The top-k condition is not applied. The dotted lines in each figure are the baselines of the dataset with the corresponding colors.

We evaluate CASD with different verification strictness in this section. Figure 2 demonstrates the performance of CASD on each data set as the threshold changes. F1 score increases with decreasing threshold for NQ while decreases for other 3 QA datasets. The ground truth is implicit in the context for each sample in NQ. A more relaxed validation condition makes the model tend to use the original statement in the context, which may be more accurate than the model's output in some cases, thus leading to better results. Therefore, for high-quality context, the threshold can be lowered to increase confidence in the context for better performance and efficiency. For both summary datasets, the overall trend of the scores decreases as the threshold decreases in the search range. But even when the threshold is set to $1e-5$, CASD still outperforms the baselines. The threshold should not be set too small to maintain the logic of the output in code completion tasks according to the observation of the results.

Table 2: Comparison of generated results with CASD under different thresholds. The question is "Who is the father of the director of film Kajraare?" The top-k requirement is not used.

| Method | Label and Predictions |
|---|---|
| Ground Truth | **Mahesh Bhatt** |
| Vanilla | Pooja Bhatt |
| CASD ($\delta$=1e-3) | Pooja Bhatt's father is **Mahesh Bhatt**. |
| CASD ($\delta$=1e-5) | Pooja Bhatt's television film Daddy was directed by her father **Mahesh Bhatt**. |
| CASD ($\delta$=1e-7) | Pooja Bhatt's television film Daddy was directed by Pooja Bhatt, starring! the director's father, played by actor Anupam Kher |

Table 2 displays the predictions under different thresholds for a sample in 2WikiMQA. Vanilla output fails to give the correct answer. However, the prediction of CASD ($\delta$=1e-3 and $\delta$=1e-5) contains the ground truth. With an appropriate threshold, CASD improves the accuracy of generation through context-aware drafting from the relevant context. When the threshold is set to 1e-5, the logic of generated content begins to lose. The prediction is completely messy when the threshold comes to 1e-7. Therefore, the threshold should be large enough to keep the model output logical and fluent.

## 4.4 IMPLEMENTATION IN PROMPT COMPRESSION SCENARIO

LLMLingua-2 (Pan et al., 2024) conducts token-level extractive text compression through a Transformer encoder, which supports adjusting the compression ratio. We evaluate the performance of CASD when using LLMlingua-2 to compress the prompt. The threshold $\delta$ is set to 0.1 for all experiments. Table 3 demonstrates the results under 3 different compression rates. CASD improves the average scores by 2.09, 1.00 and 1.3 under the three compression ratios. The performance on

Table 3: Performance of CASD on prompt compression scenario with LLaMA3.1-8B-Instruct. Prompts are compressed by LLMLingua-2 under 3 different compression ratios. $n\times$ means that we compress the context to approximately $\frac{1}{n}$ of its original length. The best results are in bold.

| Method | NQ | TQA | 2WikiMQA | HotpotQA | Multi-News | GovReport | RepoBench-P | LCC | Avg. |
|---|---|---|---|---|---|---|---|---|---|
| | F1 | F1 | F1 | F1 | ROUGE-L | ROUGE-L | Edit Sim | Edit Sim | |
| LLMLingua-2 (3×) | 25.57 | 23.51 | 35.46 | 40.94 | 12.70 | 13.70 | 57.98 | **29.63** | 29.94 |
| +CASD (Fixed) | **26.75** | **30.90** | **38.85** | **45.99** | 13.39 | 15.47 | **59.96** | 25.52 | **32.10** |
| +CASD (Mixed) | 26.58 | 30.79 | **38.85** | 45.80 | **13.47** | **15.48** | 59.89 | 25.39 | 32.03 |
| LLMLingua-2 (5×) | 24.33 | 23.51 | 30.37 | 36.67 | 12.40 | 13.22 | 58.86 | **25.49** | 28.11 |
| +CASD (Fixed) | 24.85 | **25.70** | **31.04** | 41.14 | **12.96** | **13.83** | 58.81 | 23.64 | 29.00 |
| +CASD (Mixed) | **25.00** | 25.62 | 31.00 | **41.86** | 12.88 | **13.83** | **58.93** | 23.73 | **29.11** |
| LLMLingua-2 (10×) | 22.86 | **24.88** | 23.25 | 26.81 | 11.88 | 12.11 | 55.42 | 17.81 | 24.38 |
| +CASD (Fixed) | 22.99 | 23.94 | **26.43** | **27.33** | **12.11** | **12.58** | 58.03 | **22.51** | 25.74 |
| +CASD (Mixed) | **23.02** | 23.92 | **26.43** | **27.33** | 12.08 | 12.53 | **58.40** | 22.46 | **25.77** |

different datasets varies greatly after using prompt compression. Scores on TQA, HotpotQA and LCC drop significantly while the score for RepoBench-P increases compared to baseline. A higher compression rate improves the efficiency of the prefill stage and saves the memory, but it also leads to more performance loss. CASD (Fixed) and CASD (Mixed) show comparable effects. CASD improves scores on most datasets and generally outperforms baselines in each group. The improvement is more obvious at lower compression ratios. However, for datasets like LCC, CASD reduces the score at a low compression ratio while greatly improve the performance at a high one.

These observations suggest that CASD performs effectively when the prompt provides sufficient information for the model to generate accurate answers. However, if critical information is lost due to prompt compression, the output distribution can become uncertain, reducing the reliability of the verification process. However, in the case of inferior prompt quality (e.g., LCC in the 10× group), CASD simply pieces together some fragments from the context based on the current prompt, which may bring gains in the score.

## 5 ANALYSIS

### 5.1 INFERENCE EFFICIENCY

We test the mean acceptance length (MAL) and throughput of CASD with LLaMA3.1-8B-Instruct on NQ, GovReport and LCC (one dataset for each task). MAL is defined as the average accepted tokens per decoding step. If all the draft tokens are rejected, the acceptance length is 1 since we sample the token with the biggest probability in the next token distribution. The threshold is set to 1e-5 for NQ and 0.1 for the other two datasets to study the efficiency when the output has high quality. Table 4 shows the average results of three runs tested on a single A100-PCIE-40GB GPU. The mean accuracy length on 3 datasets is greater than 2. CASD achieves a speed-up ratio between 1.63 and 1.99 on the standard setting. However, although the input length is shortened, the speed-up ratio drops to 1.01~1.85 on the prompt compression setting. Prompt compression reduces the time overhead of the prefill phase. However, the lower input quality influences the effectiveness of the verification method, result-

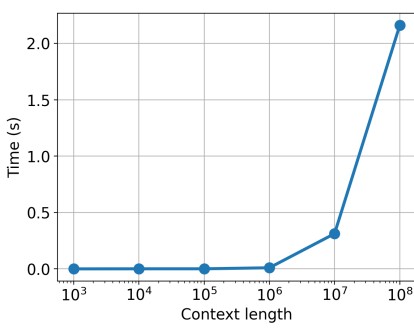

Figure 3: Draft retrieval time overhead under different context lengths.

ing in a smaller mean acceptance length. Judging from experimental results, though combining CASD and prompt compression methods can significantly reduce the memory overhead, it does not necessarily increase throughput.

Table 4: Inference efficiency of CASD under standard and prompt compression settings. Lingua represents LLM-Lingua2. MAL and TPS indicate mean acceptance length and tokens per second, respectively. The "Speed-up" is the ratio of the method's TPS to the TPS of vanilla.

| Method | NQ | | | GovReport | | | LCC | | |
|---|---|---|---|---|---|---|---|---|---|
| | MAL | TPS | Speed-up | MAL | TPS | Speed-up | MAL | TPS | Speed-up |
| Vanilla | 1.00 | 24.61 | 1.00 | 1.00 | 28.45 | 1.00 | 1.00 | 27.02 | 1.00 |
| **CASD** | 3.10 | 48.94 | 1.99 | 2.02 | 46.48 | 1.63 | 2.58 | 48.58 | 1.80 |
| **CASD**+Lingua(3×) | 1.22 | 28.20 | 1.15 | 1.24 | 31.04 | 1.09 | 2.08 | 49.92 | 1.85 |
| **CASD**+Lingua(5×) | 1.21 | 30.31 | 1.23 | 1.16 | 29.41 | 1.03 | 1.74 | 43.29 | 1.60 |
| **CASD**+Lingua(10×) | 1.21 | 30.46 | 1.24 | 1.06 | 28.64 | 1.01 | 1.17 | 29.81 | 1.10 |

In addition, we test the draft retrieval time for context of different lengths, which is shown in Figure 3. The retrieval time is between 0 and 100 milliseconds for context within 1M tokens, which has little impact on end-to-end inference efficiency.

## 5.2 CASE STUDY

Table 5: An example of generation with CASD on extractive QA task. The question is "where does the term dog and pony show come from". There are two candidates for the ground truth. We omit some irrelevant part for brevity. The highlighted text of the corresponding mark shows the part where CASD output is highly similar to the ground truth, which is also given in the relevant context.

**Ground Truth:**

[" Dog and pony show " is a colloquial term which has come to mean a ♠ highly promoted, often over-staged performance, presentation, or event designed to sway or convince opinion for political, or less often, commercial ends. Typically, the term is ♣ used in a pejorative sense to connote disdain , jocular lack of appreciation, or distrust of the message being presented or the efforts undertaken to present it.,

The term was originally used ♦ in the United States in the late - 19th and early - 20th centuries to refer to small traveling circuses that toured through small towns and rural areas. The name derives from the common use of performing dogs and ponies as the main attractions of the events. Performances were generally held in open - air arenas, such as race tracks or public spaces in localities that were too small or remote to attract larger, more elaborate performers or performances.(....)]

**Vanilla Output:**

The term "dog and pony show" is believed to have originated in the United States in the late 19th and early 20th centuries. It was originally used to refer to small traveling circuses that toured through small towns and rural areas, featuring performing dogs and ponies as the main attractions. These circuses were often over-hyped and over-staged, with the goal of convincing or entertaining the audience rather than providing a genuine performance. Over time, the term "dog and pony show" came to be used more broadly to describe any highly promoted or over-staged event or presentation, often with a negative connotation.

**CASD Output:**

The term "dog and pony show" originated ♦ in the United States in the late 19th and early 20th centuries to refer to small traveling circuses that toured through small towns and rural areas. The name derives from the common use of performing dogs and ponies as the main attractions of the events. These performances were generally held in open-air arenas, such as race tracks or public spaces in localities that were too small or remote to attract larger, more elaborate performers or performances.

The term "dog and pony show" was originally ♣ used in a pejorative sense to connote disdain, jocular lack of appreciation, or distrust of the message being presented or the efforts undertaken to present it. Over time, the term has come to be used more broadly to refer to any ♠ highly promoted or over-staged performance, presentation, or event designed to sway or convince opinion for political or commercial ends.

To intuitively show the effect of CASD, we pick an example in the NQ dataset, which is displayed in Table 5. The ground truth can be found in the provided context. The F1 score of vanilla output

and CASD output is 46.09 and 59.21, respectively. The output of CASD demonstrates high similarity with the ground truth. Although CASD begins with the same few tokens as vanilla output, it accurately generates the key points in the ground truth.

Even if the answer is given in the prompt, LMs would paraphrase in their own words based on the habits acquired through pre-training. This characteristic may sometimes cause the model's response to deviate from the given facts. CASD adds an interface to leverage the original information of the context on the decoding side through context-aware speculative decoding, making the model more inclined to use the original statements in the context to answer questions. Especially for some specific nouns, the expression in the context will be more precise. Therefore, CASD improves the accuracy of the generated answer.

For the NQ dataset, where the ground truth is contained in the relevant context, simply piecing together the original sentence in the input may also improve the F1 score. Therefore, we attempt to use the GPT-4 model (OpenAI et al., 2024) to analyze the quality of the generation. GPT-4 also believes that the output of CASD is superior to the vanilla output. In terms of content, CASD's answer more accurately captures the origin of "dog and pony show" and the changes in its meaning. Moreover, the CASD output covers the background, origin and evolution of the word while being more organized and concise.

## 5.3 Ablation Study

Table 6: Results of the ablation study. W/o threshold means using the top-k condition only for the verification phase.

| Method | NQ | TQA | 2WikiMQA | HotpotQA | Multi-News | GovReport | RepoBench-P | LCC | Avg. |
|---|---|---|---|---|---|---|---|---|---|
| | F1 | F1 | F1 | F1 | ROUGE-L | ROUGE-L | Edit Sim | Edit Sim | |
| CASD (Mixed) | 33.69 | 92.62 | 43.95 | 52.46 | 14.83 | 20.06 | 56.63 | 69.46 | 47.96 |
| w/o threshold | 37.47 | 92.64 | 42.84 | 50.20 | 15.16 | 19.31 | 49.86 | 60.80 | 46.04 |

We conduct the ablation study in this section. We evaluate the performance of CASD with the top-k verification condition only, which is compared with CASD (Mixed). Results are shown in Table 6. Using the top-k condition alone decreases the scores for most datasets, especially for RepoBench-P and LCC. NQ makes a difference since it fits more relaxed acceptance conditions. On the other hand, removing the top-k condition (CASD (Fixed)) decreases the average score, which is already shown in Table 1. In general, it is beneficial to combine the two verification conditions.

## 6 Discussion

In this paper, we proposed CASD, a context-aware speculative decoding algorithm with conditional verification, which enhances the generation accuracy of LMs at the token level. We explored the potential for leveraging the input context in the decoding stage. CASD increases the probability that the model adopts the tokens in the relevant context, thus making better use of the given information. It does not introduce additional parameters and training, thus can be conveniently applied to pre-trained LMs. CASD outperforms the baseline in experiments on 8 datasets in terms of generation performance and efficiency. Implementation of CASD in prompt compression scenarios also benefits the performance. CASD currently uses heuristic verification rules. If the verification strictness can be adaptively set according to different inputs, the versatility of this method would be further improved. We would like to explore adaptive verification conditions at the request level or token level in future work.

**Limitations** CASD accepts the longest verified draft in each decoding step to pursue inference efficiency. However, this generation paradigm conflicts with the sampling methods commonly used in autoregressive models, such as greedy sampling and temperature sampling. Besides, if the input context contain harmful information, using CASD may lead the model to incorporate it into its responses. Therefore, in practical applications, CASD requires additional safety alignment to ensure the security of the language model.

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

## A    EXPERIMENTAL DETAILS

Table 7 shows the prompts used in the experiments in this paper. The instructions or questions are unified for some datasets. So there is no separate question for each sample.

## B    SUPPLEMENTARY RESULTS

Table 8 shows the specific performance of casd on each data set under different thresholds. Figure 4 displays the curve of MAL as the verification threshold decreases.

Table 7: Prompts used in experiments for all datasets. $\{context\}$ represents the relevant context for each sample. $\{question\}$ indicates the query or instruct.

| Dataset | Prompt |
|---|---|
| NQ | Answer the Question based on the given context. \n\n Context:$\{context\}$ \n\n Question:$\{question\}$ \n\n Answer: |
| TQA | Answer the question based on the given passage. Only give me the answer and do not output any other words. The following are some examples. \n\n $\{context\}$ \n\n $\{question\}$ |
| 2WikiMQA | Answer the question based on the given passages. Only give me the answer and do not output any other words. \n\n The following are given passages.\n $\{context\}$ \n\n Answer the question based on the given passages. Only give me the answer and do not output any other words. \n\n Question:$\{question\}$ \n Answer: |
| HotpotQA | Answer the question based on the given passages. Only give me the answer and do not output any other words. \n\n The following are given passages.\n $\{context\}$ \n\n Answer the question based on the given passages. Only give me the answer and do not output any other words. \n\n Question:$\{question\}$ \n Answer: |
| Multi-News | You are given several news passages. Write a one-page summary of all news. \n\n News:\n $\{context\}$ \n\n Now, write a one-page summary of all the news. \n\n Summary: |
| GovReport | You are given a report by a government agency. Write a one-page summary of the report. \n\n Report:\n $\{context\}$ \n\n Now, write a one-page summary of the report. \n\n Summary: |
| RepoBench-P | Please complete the code given below. \n $\{context\}$ $\{question\}$ Next line of code:\n |
| LCC | Please complete the code given below. \n $\{context\}$ Next line of code:\n |

Table 8: Performance of CASD under different thresholds. Best results are in bold.

| Threshold | NQ | TQA | 2WikiMQA | HotpotQA | Multi-News | GovReport | RepoBench-P | LCC | Avg. |
|---|---|---|---|---|---|---|---|---|---|
| | F1 | F1 | F1 | F1 | ROUGE-L | ROUGE-L | Edit Sim | Edit Sim | |
| 0.1 | 33.51 | 92.62 | **44.67** | **51.72** | 14.97 | **20.09** | **56.31** | **69.24** | **47.89** |
| 0.01 | 35.51 | **92.75** | 38.29 | 49.74 | **15.14** | 19.89 | 48.98 | 61.50 | 45.23 |
| 1e-3 | 40.02 | 92.51 | 31.24 | 42.93 | 15.12 | 19.31 | 43.31 | 55.56 | 40.63 |
| 1e-4 | 42.83 | 91.83 | 28.02 | 36.36 | 14.69 | 18.52 | 40.19 | 52.75 | 40.65 |
| 1e-5 | **43.78** | 88.96 | 26.31 | 33.02 | 14.06 | 17.74 | 38.43 | 49.58 | 38.99 |

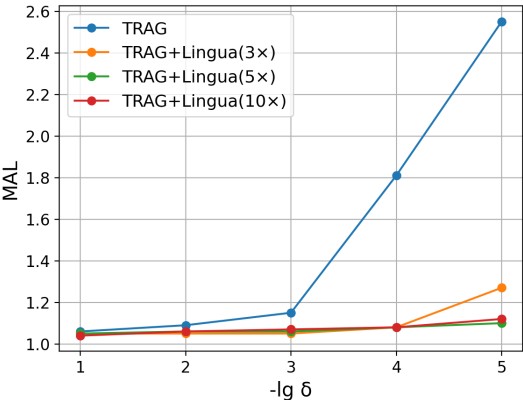

Figure 4: Mean acceptance lenghth under different thresholds.

