# OpenReview forum: "CASD: Enhancing Generation Accuracy via Context-Aware Speculative Decoding"
_ICLR.cc/2025/Conference — ICLR 2025 Conference Withdrawn Submission_

### Official Review · Reviewer_xHwq · 2024-10-30

**Soundness:** 2
**Presentation:** 2
**Contribution:** 1
**Rating:** 3
**Confidence:** 4

**Summary:**

This paper proposes a modified version of the REST algorithm for speculative decoding called : 1) they use the input context as the pool for draft retrieval. 2) they decide whether to accept the draft by comparing the conditional probability with a predefined threshold. Experimental results show that the proposed method can achieve better accuracy on question answering, summarization and code completion, and a speed-up ratio from 1.63 to 1.99.

**Strengths:**

1. It is reasonable to use the input context as the draft pool as the retrieval when the input context is long and the task includes extraction or summarization.
2. Experimental results show that the method can improve the accuracy while speeding up inference, which is not possible for previous methods of speculative decoding.

**Weaknesses:**

1. Overall the method is a minor modification of REST, simply replacing the external database with the input context. It is only a new way to build the retrieval database.
2. The modification is less general than the original REST algorithm, since it only applies to tasks whose prompts are long and the answers are likely to appear in the prompts. When the input prompts are short, the modification will not work.
3. Some important experiment results are missing. No comparison with traditional speculative decoding (with a small LM) on speedup ratios is provided. To ablate the effect of conditional verification with the threshold, the method should be compared with the acceptance method of REST, not top-k verification.

**Questions:**

1. In section 5.1, the threshold on NQ is set to 1e-5, while the threshold on the other two datasets is 0.1. What is the reason here? I note that in section 4.1 the threshold is 0.1 on all the datasets. Also, the speed-up ratio on NQ is the highest. Is it because the threshold is lower than that of other datasets?
2. On line 369 the word "mean accuracy length" should be "mean acceptance length".

---

### Official Review · Reviewer_GwK7 · 2024-10-31

**Soundness:** 1
**Presentation:** 1
**Contribution:** 2
**Rating:** 3
**Confidence:** 4

**Summary:**

The paper presents Context-Aware Speculative Decoding (CASD), a decoding method to improve LLM performance in tasks involving extensive input contexts. The method is simple, involving three steps: (1) Retrieve high-quality drafts from the contexts relevant to the input prompt and (2) Get the next token distributions for all tokens in the draft; (3) Accept the longest draft if tokens there match the tokens sample from the LLM at a certain probability threshold.

**Strengths:**

The method shows promising improvements and speedups.

**Weaknesses:**

1. **Limited novelty**: This paper primarily focuses on draft retrieval and employs a probability threshold to accept or reject tokens, which I consider as an incremental contribution.

2. **Limited soundness**: Unlike original Speculative Decoding, CASD uses the original model as the draft model to compute probabilities, which seems like a backward step. The draft model in Speculative Decoding plays a crucial role in speeding up the method; relying on the original model for probabilities may yield only minor speed gains. Additionally, CASD may violate the traditional autoregressive decoding style since it accepts the longest draft, which may cause unwanted text behavior.

3. **Limited baseline comparison**: The paper does not compare CASD with other prompting baselines such as the original SD. In addition, its experiments are conducted with only LLama.

4. **Threshold sensitivity**: The results indicate that performance varies significantly with different threshold settings.

5. **Limited writing**: The writing can be significantly improved. For example, the details of prompt compression are limited. It's unclear why using  LLMLingua-2 and it's unclear why it works.

**Questions:**

See weaknesses

---

### Official Review · Reviewer_e3hR · 2024-11-03

**Soundness:** 2
**Presentation:** 3
**Contribution:** 2
**Rating:** 3
**Confidence:** 4

**Summary:**

This paper proposes a method called CASD (Context-Aware Speculative Decoding) for leveraging input context effectively and efficiently. This method uses the original text as the draft source which requires no additional training or draft models. In this paper, they also propose to replace the strict verification in conventional speculative decoding with conditional verification, where pieces of the relevant context are verified and accepted at the token level. They conduct experiments on 8 datasets and show better accuracy and a speed-up ratio of 1.99.

**Strengths:**

* CASD is a plug-and-play method that requires no additional training or draft models, making it easy to implement. It can also be effectively combined with prompt compression methods, which are used to reduce the computational burden of long contexts.

* The proposed conditional verification allows for flexibility in balancing context reliance and generation quality.  Experimental results on various datasets show that the generation scores of CASD are improved.

* Besides improved accuracy, CASD also reduces inference latency.It achieves a speed-up ratio of up to 2 times compared to traditional autoregressive decoding.

**Weaknesses:**

* One limitation of CASD is its generalizability to all tasks. While it excels in tasks that benefit from direct context, its effectiveness may be limited in tasks requiring abstractive or creative generations.

* As stated in the paper, CASD conflicts with common sampling techniques in autoregressive models, which makes it less practical for many scenarios that require multiple generations per sample.

* Although the experiments are conducted on multiple datasets, the baseline is too limited, which is vanilla decoding of LlaMa3.1-8B-instruct model only. While it’s understandable to skip decoding methods that require additional training or draft models, there are other training-free speculative decoding methods available to compare with. Comparisons with these alternative approaches, such as REST, are crucial for a more comprehensive evaluation.

* The choice of the probability threshold  for accepting draft tokens during verification significantly impacts CASD's performance. The sources acknowledge that a lower threshold can enhance accuracy with reliable contexts but might compromise fluency. However, the experiments primarily rely on fixed or manually tuned thresholds, which lack generalizability and require prior knowledge of the dataset characteristics. It would be better if the threshold could be determined automatically by a generalizable heuristic rule.

* The setting of the vanilla decoding is unclear anywhere in this paper, i.e., temperature, sampling sizes, maximum context length and maximum generation token length. These can affect the model's performance and thus should be described in the experiments section.

**Questions:**

* One main contribution of the proposed method is improving the generation accuracy.  In ths paper, only one model is evaluated.  How does the base model's performance affect the improvement?

* In table 1, Fixed and Mixed methods have different performance trend on different tasks. Which one is recommended and more generalizable?

---

### Official Review · Reviewer_vYD8 · 2024-11-04

**Soundness:** 2
**Presentation:** 3
**Contribution:** 2
**Rating:** 3
**Confidence:** 4

**Summary:**

This paper introduce a novel speculative decoding strategy named CASD. CASD incorporates context-aware speculative decoding with a conditional verification approach, allowing LLMs to reuse relevant context at the token level without additional training or models. CASD achieves superior performance across tasks like question answering, summarization, and code completion, demonstrating a +3.3 increase in average generation scores and a nearly 2x increase in inference speed.

**Strengths:**

1. Enhances generation accuracy and nearly doubles inference speed without additional training or models.
2. Works well with context compression.

**Weaknesses:**

1. The difference between the proposed method and REST [1] is minimal, same exact retrieval algorithm is used; the general database in REST could also be replaced by the input context.
2. Training-free / model-free speculate decoding baselines (e.g. [2]) is not compared.
3. Different templates is need for different datasets. It's not clear whether CASD is suitable for the most common chatbots scenarios.
4. This method tends to make the output distribution more inclined to restate the input context, which is advantageous for extractive QA tasks where the answer is present in the input. However, its impact on more general scenarios remains unclear.

[1] REST: Retrieval-Based Speculative Decoding
[2] Inference with Reference: Lossless Acceleration of Large Language Models

**Questions:**

1. More baselines can be introduced.
2. How do you choose the 0.1 threshold for all experiments?

---

### Note · Authors · 2024-11-18

I have read and agree with the venue's withdrawal policy on behalf of myself and my co-authors.